

# The tuberculocidal activity of polyaniline and functionalised polyanilines

Julia Robertson[1], James Dalton[1], Siouxsie Wiles[1,2], Marija Gizdavic-Nikolaidis[3] and Simon Swift[1]

[1] Department of Molecular Medicine and Pathology, University of Auckland, Auckland, New Zealand
[2] Bioluminescent Superbugs Lab, Department of Molecular Medicine and Pathology, University of Auckland, Auckland, New Zealand
[3] School of Chemistry, University of Auckland, Auckland, New Zealand

## ABSTRACT

Tuberculosis is considered a leading cause of death worldwide. More than 95% of cases and deaths occur in low- and middle-income countries. In resource-limited countries, hospitals often lack adequate facilities to manage and isolate patients with infectious tuberculosis (TB), relying instead on personal protective equipment, such as facemasks, to reduce nosocomial transmission of the disease. Facemasks impregnated with an antimicrobial agent may be a cost-effective way of adding an extra level of protection against the spread of TB by reducing the risk of disease transmission. Conducting polymers, such as polyaniline (PANI), and their functionalised derivatives are a novel class of antimicrobial agents with potential as non-leaching additives to provide contamination resistant surfaces. We have investigated the antimicrobial action of PANI and a functionalised derivative, poly-3-aminobenzoic acid (P3ABA), against mycobacteria and have determined the optimal treatment time and concentration to achieve significant knockdown of *Mycobacterium smegmatis* and *Mycobacterium tuberculosis* on an agar surface. Results indicated that P3ABA is a potential candidate for use as an anti-tuberculoid agent in facemasks to reduce TB transmission.

Corresponding author
Simon Swift, s.swift@auckland.ac.nz

## INTRODUCTION

Tuberculosis (TB) ranks alongside the human immunodeficiency virus (HIV) as a leading cause of death worldwide (*WHO, 2015*). It has been estimated that in 2014 9.6 million people contracted, and 1.5 million people died from, TB (*WHO, 2015*). While the global burden of TB has been decreasing over the last decade, more needs to be done as most deaths from TB are preventable and infections due to drug resistant TB are on the rise (*WHO, 2015*). More than 95% of cases and deaths occur in low- and middle-income countries (LMICs), notably in South-East Asia and African regions (*WHO, 2015*). Multidrug resistant tuberculosis (MDR-TB), caused by *Mycobacterium tuberculosis* that is resistant to both isoniazid and rifampin, is a threat to TB control (*WHO, 2015*; *Menon, 2013*). It is estimated that 480,000 people (5% of cases worldwide) developed MDR-TB in 2014 while an estimated 190,000 people (13% of deaths worldwide) died from MDR-TB infection (*WHO, 2015*).

MDR-TB is difficult to cure as treatment is required for longer periods of time with more costly drugs with toxic side effects (*WHO, 2015*).

Conducting polymers (CPs) and their functionalised derivatives are a novel class of antimicrobial agents that may be used as an additive to provide contamination resistant surfaces. PANI and its functionalised derivatives (fPANIs) comprise a widely studied class of conducting polymers (*Dhand et al., 2011*). Utilisation of PANI for potential applications is restricted because of its insolubility in common solvents, which renders it difficult to process (*Gizdavic-Nikolaidis et al., 2011b*; *Pandey, Annapoorni & Malhotra, 1993*). fPANIs are easily and inexpensively synthesised using substituted aniline monomers, which improves the solubility, and thus processability, of the resulting polymer (*Gizdavic-Nikolaidis et al., 2011b*; *Pandey, Annapoorni & Malhotra, 1993*). PANI and fPANIs have good thermal stability and have been incorporated as melt blends in plastics (*Nand et al., 2013*) and electrospun in nanofibers (*Gizdavic-Nikolaidis et al., 2010a*) where antimicrobial activity is retained. These novel antimicrobial agents have broad spectrum activity against gram-negative and gram-positive bacteria, but activity against mycobacteria has not been reported (*Gizdavic-Nikolaidis et al., 2011a*; *Shi et al., 2006*). Preliminary studies have suggested fPANIs have a multifunctional mechanism of action that act, in part, via perturbation of aerobic energy metabolism to cause lethal oxidative damage (*Gizdavic-Nikolaidis et al., 2011a*). We hypothesise that there will be activity against mycobacteria given that they are obligate aerobes and are sensitive to agents that interfere with aerobic energy metabolism (*Hards et al., 2015*; *Leibert, Danckers & Rom, 2014*; *Koul et al., 2014*). Any role for conduction in antimicrobial activity is unclear, P3ABA is $10^3$–$10^4$ less conductive than PANI (*Gizdavic-Nikolaidis et al., 2010a*; *Gizdavic-Nikolaidis et al., 2010b*).

In 2006, the WHO developed the 'Stop TB strategy' with the aim of diminishing the global TB burden (*WHO, 2015*). One vital component of this strategy is improved infection control measures in health facilities to reduce transmission, where effective use of personal protective equipment (PPE) is necessary (*WHO, 2015*; *Menon, 2013*; *WHO, 2014*; *Buregyeya et al., 2013*). However, healthcare facilities in LMICs rarely implement recommended TB infection control measures due to financial and resource constraints (*Menon, 2013*; *Gonzalez-Angulo et al., 2013*; *Buregyeya et al., 2013*; *Kompala, Shenoi & Friedland, 2013*) and health care workers (HCWs) are at greater risk of contracting drug-resistant TB as a result (*O'Donnell et al., 2010*; *Joshi et al., 2006*).

Facemasks fabricated from materials impregnated with an antimicrobial agent are a potential cost-effective strategy to protect against the spread of TB (*Yang et al., 2011*). Antimicrobial facemasks may decrease the risk of TB transmission that occurs due to reuse of contaminated facemasks or reuse of facemasks that have been structurally compromised during decontamination (*Menon, 2013*; *Yang et al., 2011*; *Chughtai et al., 2015*). Contamination resistant facemasks are a realistic infection control measure for resource-constrained settings as they would not need to be replaced as frequently as disposable masks. This would reduce the overall expense of providing facemasks to HCWs and could increase facemask availability.

In this study we have investigated the antimicrobial action of PANI and an fPANI, P3ABA, against mycobacteria genetically tagged for bioluminescence as a marker for

cellular viability. Bioluminescence is the production of light via a luciferase catalysed reaction (*Andreu, Zelmer & Wiles, 2011*). As tagged cells only produce a signal when alive, bioluminescence is an excellent reporter to rapidly assay for antimicrobial compounds, non-destructively and in real-time, in microtitre plate formats using a luminometer, or *in vivo* using sensitive imaging equipment (*Andreu et al., 2013*; *Andreu et al., 2012*). We determined the optimal treatment time and concentration to achieve a $>10^3$-fold knockdown for bioluminescently-tagged strains of *M. smegmatis* and *M. tuberculosis* inoculated onto a solid surface. Results indicated that P3ABA is a potential candidate for use as an antimicrobial agent incorporated into materials and PPE in the patient environment to reduce TB transmission.

## METHODS AND MATERIALS

### Bacterial strains and growth conditions

*M. smegmatis* ATCC 700084 was tagged with an integrating plasmid (pMVhspLux-ABG13CDE) containing the bacterial luciferase (*lux*) operon and designated BSG200 using a standard method (*Wiles et al., 2005*; *Andreu et al., 2010*). *M. tuberculosis* reference strain ATCC 27294 and *M. tuberculosis* clinical isolate Rangipo N were similarly tagged and designated BSG001 and BSG002, respectively (*Colangeli et al., 2014*; *Wang et al., 2016*). All strains were grown at 37 °C, with 200 rpm agitation where appropriate. The University of Auckland Institutional Biological Safety Committee approved the construction and use of genetically modified risk group 2 mycobacteria (GMO11-UA007). The New Zealand Environmental Protection Agency approved the construction and use of genetically modified risk group 3 *Mycobacterium tuberculosis* (APP201346).

### Media and chemicals

All strains were cultured in Middlebrook 7H9 broth supplemented with 10% Albumin Dextrose Catalase (ADC) and 0.5% glycerol (subsequently referred to as supplemented 7H9) or on Middlebrook 7H11 agar supplemented with 10% Oleic Albumin Dextrose Catalase (OADC) and 5% glycerol (subsequently referred to as supplemented 7H11). Bacterial growth media were obtained from Fort Richard, Auckland. For the starting inocula, 0.05% Tween 80 (Sigma Aldrich) was added to the supplemented 7H9 to avoid cellular agglomeration. To generate the experimental inocula, cells were diluted in supplemented 7H9 without tween 80. PANI and P3ABA were synthesised via chemical oxidation of aniline and 3-aminobenzoic acid monomers, respectively (*Gizdavic-Nikolaidis et al., 2011a*).

### Antibacterial action of surface incorporated PANI and P3ABA against *M. smegmatis*

Aliquots (200 μl) of supplemented 7H11 agar containing 8% PANI, 10% PANI, 1% P3ABA, 2% P3ABA or 3.5% P3ABA were added to wells of a black 96-well plate (Perkin Elmer) and allowed to set. In each well the surface of the agar was inoculated with $\sim10^4$ CFU *M. smegmatis* BSG200 in 10 μl and incubated at 37 °C. At 15 min, 30 min and 120 min *M. smegmatis* BSG200 cells were recovered by addition of 190 μl supplemented 7H9 onto the agar, pipetting up and down 8-times, and transfer of the broth to a fresh 96-well plate.

This plate was incubated at 37 °C and bioluminescence was measured using a Victor$^{TM}$ $X$ light luminescence plate reader (2030-0010) (Perkin Elmer) at 0 h and 24 h incubation. Uninoculated wells were measured to generate background luminescence readings. The experiment was performed in triplicate using independent cultures.

### Antibacterial action of surface incorporated PANI and P3ABA against *M. tuberculosis*

The method developed for *M. smegmatis* was adapted to examine the tuberculocidal action of surface incorporated PANI and P3ABA. *M. tuberculosis* BSG001 or *M. tuberculosis* BSG002 were inoculated onto agar in black 96-well plates and rescued at the stated time points. The recovered cells were incubated at 37 °C . The level of bioluminescence was measured after approximately 21 days using a Victor$^{TM}$ $X$ light luminescence plate reader and growth was assessed visually. The experiments were done in triplicate using independent cultures.

### Statistical analyses

Statistical analysis was performed using GraphPad Prism software version 6 (GraphPad Software, Inc.). Analysis of the effect of PANI and P3ABA treatment was based on the bioluminescence measurement 24 h post-rescue for *M. smegmatis* and 21 days post-rescue for *M. tuberculosis*. The Friedman test was used to detect differences in antimicrobial treatments utilising a 5% level of statistical significance. To determine which PANI or P3ABA concentration could achieve significant knockdown with the shortest exposure time, where 15, 30 and 120 min challenges were made, Dunn's multiple comparison post-hoc test was used.

## RESULTS

### Antibacterial action of surface incorporated PANI and P3ABA against *M. smegmatis* BSG200

To investigate the activity of PANI and P3ABA against *M. tuberculosis*, the agar surface decontamination assay was first optimised using *M. smegmatis* BSG200. *M. smegmatis* is used as a safer and faster growing surrogate for the highly pathogenic *M. tuberculosis* (*Chaturvedi et al., 2007*). The surface of agar with 10% and 8% PANI reduced bioluminescence measurements from *M. smegmatis* BSG200 to background levels after 30 min and 120 min exposure times, respectively (Fig. 1A). The activity of PANI was statistically significant (Friedman test, $P$ value: 0.0033), with both 10% and 8% PANI treatment differing significantly from untreated cells at the 120 min time point (Dunn's multiple comparison post-hoc test).

Agar containing 2% P3ABA and 3.5% P3ABA reduced bioluminescence levels from surface-inoculated *M. smegmatis* BSG200 cells to that of uninoculated agar following a 15 min treatment (Fig. 1B). Agar containing 1% P3ABA demonstrated variable activity with consistent knockdown of surface-inoculated *M. smegmatis* BSG200 cells occurring after 30 min exposure (Fig. 1B). The overall antimicrobial action of 3.5% P3ABA was statistically significant (Friedman test, $P$ value: 0.0044). Dunn's multiple comparison test confirmed

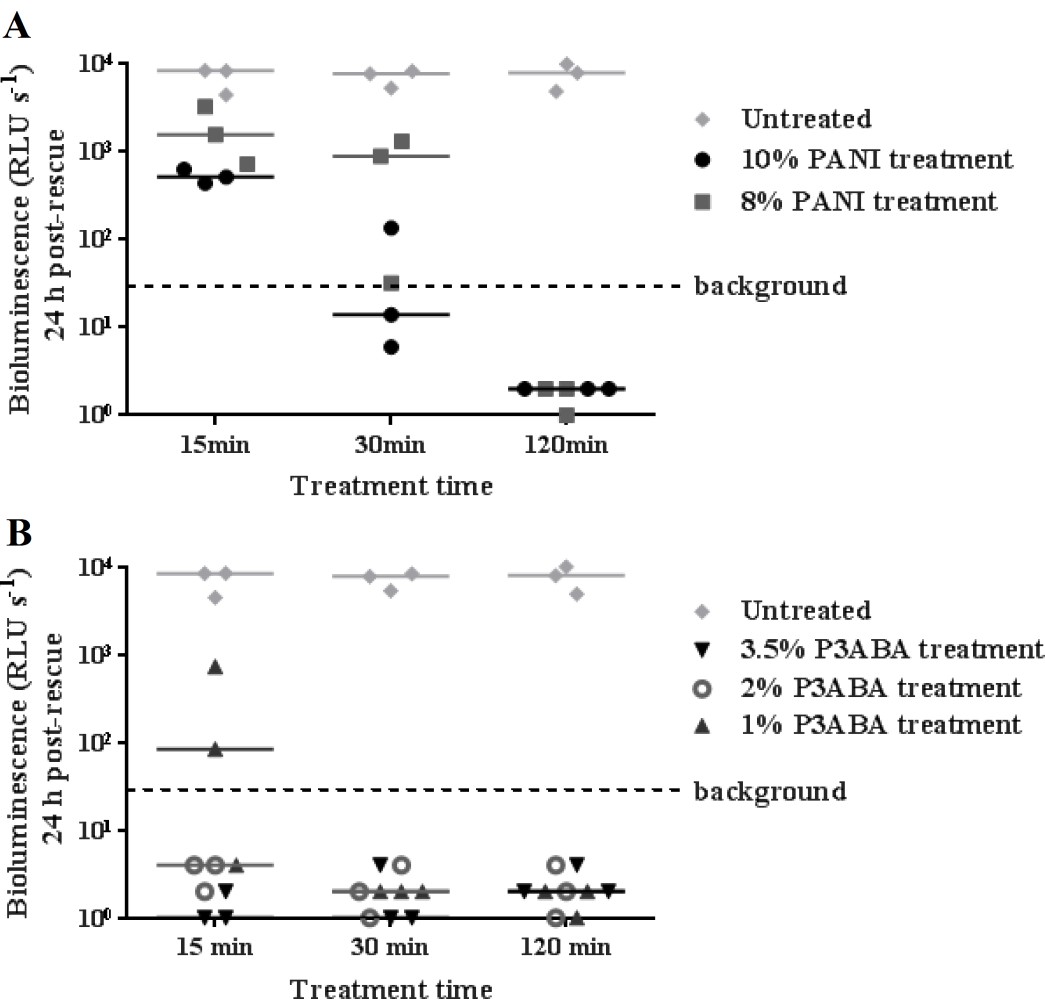

**Figure 1  Antimicrobial action of surface incorporated PANI and P3ABA against *M. smegmatis* BSG200.** (A) *M. smegmatis* BSG200 cells were exposed to the surface of agar containing 10% PANI (closed circles) and 8% PANI (squares). (B) *M. smegmatis* BSG200 cells were exposed to the surface of agar containing 3.5% P3ABA (downward triangles), 2% P3ABA (open circles) and 1% P3ABA (upward triangles). Following 15 min, 30 min and 120 min treatments, *M. smegmatis* BSG200 cells were recovered by addition of supplemented 7H9 and incubation at 37 °C in a fresh 96-well plate. The vertical axis represents the median bioluminescence measurements (given as relative light units [RLU] per second) from recovered cells grown for 24 h. The dashed line denotes background luminescence of uninoculated wells.

the significant activity of 3.5% P3ABA in a 15 min treatment time. Having established the protocol for fast-growing, non-pathogenic mycobacteria, testing was extended to the slow-growing human pathogen *M. tuberculosis.*

## Antibacterial action of surface incorporated PANI and P3ABA against *M. tuberculosis* BSG001

The activity of the polymers against *M. tuberculosis* was first investigated with the tagged laboratory reference strain *M. tuberculosis* BSG001. Both 10% and 8% PANI agar did not mediate a reduction of *M. tuberculosis* BSG001 after treatment for 120 min (Fig. 2A). This

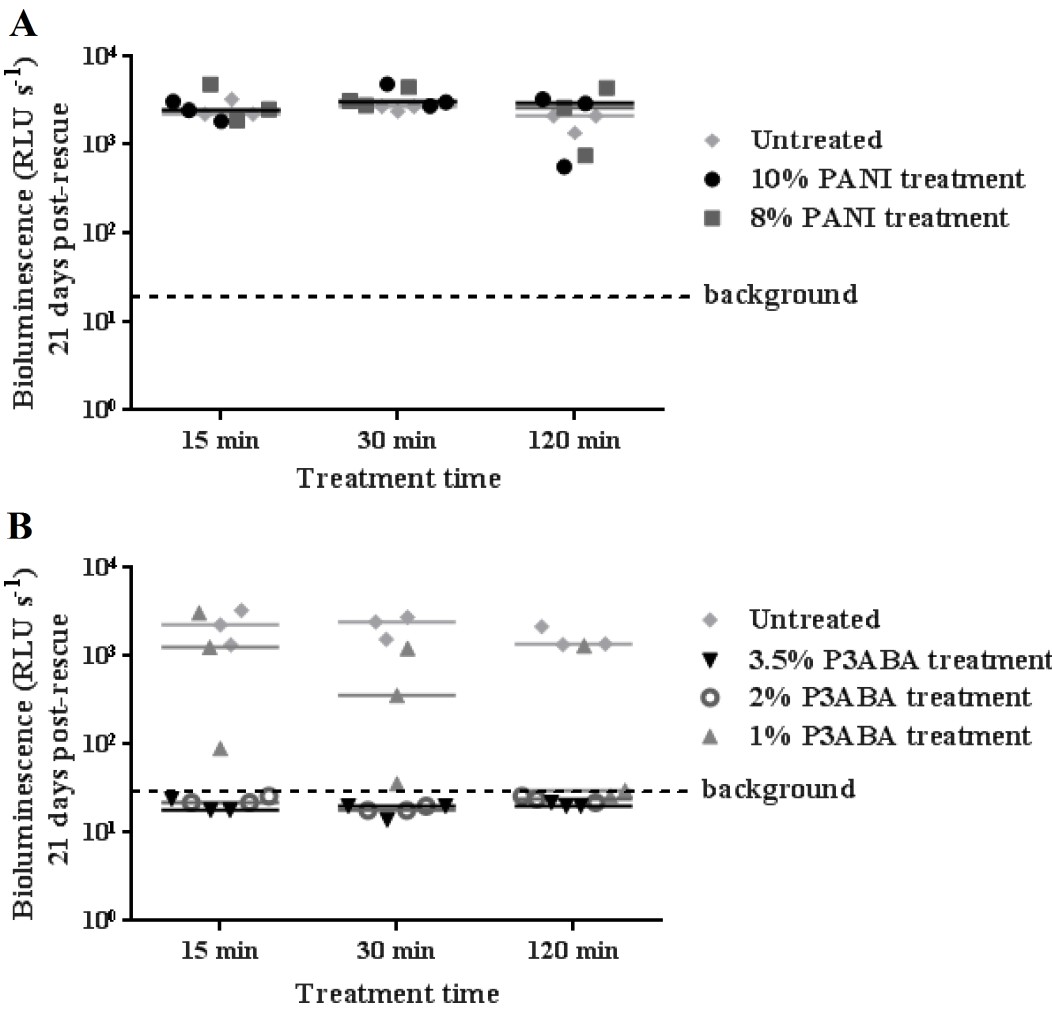

**Figure 2** **Antimicrobial action of surface incorporated PANI and P3ABA against *M. tuberculosis* BSG001.** (A) *M. tuberculosis* BSG001 cells were exposed to the surface of agar containing 10% PANI (closed circles) and 8% PANI (squares). (B) *M. tuberculosis* BSG001 cells were exposed to the surface of agar containing 3.5% P3ABA (downward triangles), 2% P3ABA (open circles) and 1% P3ABA (upward triangles). Following 15 min, 30 min and 120 min treatments, *M. tuberculosis* BSG001 cells were recovered by addition of supplemented 7H9 and incubation at 37 °C in a fresh 96-well plate. The vertical axis represents the median bioluminescence measurements (given as relative light units [RLU] per second) from recovered cells grown for approximately 21 days. The dashed line denotes background luminescence of uninoculated wells.

is in contrast to the action of PANI against *M. smegmatis* BSG200 (Fig. 1A). A 15 min exposure of surface-inoculated *M. tuberculosis* BSG001 cells to 2% and 3.5% P3ABA agar reduced bioluminescence levels to that of background levels (Fig. 2B). The action of 1% P3ABA agar against surface-inoculated *M. tuberculosis* BSG001 was variable with most cells being killed after 120 min treatment (Fig. 2B). Knockdown of *M. tuberculosis* BSG001 following 30 min exposure to the agar surface containing 3.5% P3ABA was statistically significant (Friedman test, *P* value: 0.0015; Dunn's multiple comparison post-hoc test).

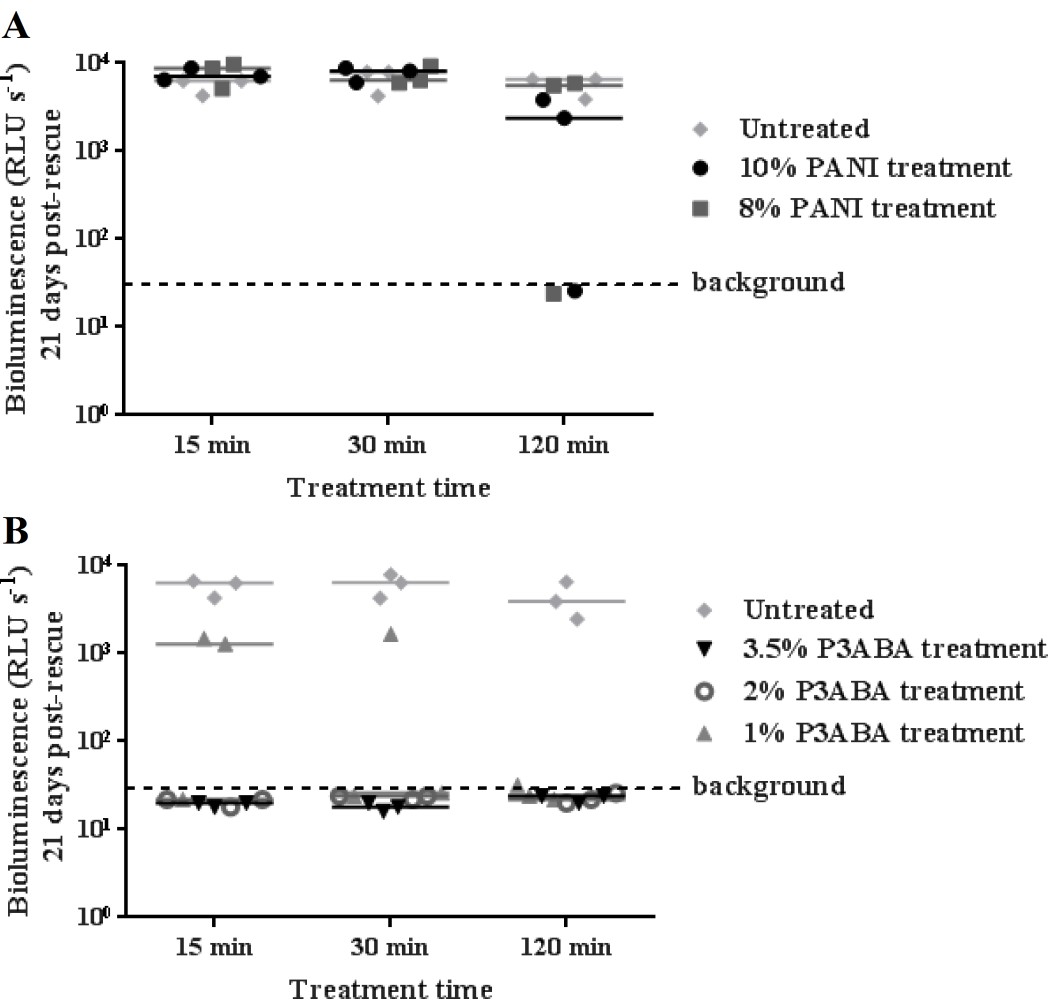

**Figure 3** **Antimicrobial action of surface incorporated PANI and P3ABA against *M. tuberculosis* BSG002.** (A) *M. tuberculosis* BSG002 cells were exposed to the surface of agar containing 10% PANI (closed circles) and 8% PANI (squares). (B) *M. tuberculosis* BSG002 cells were exposed to the surface of agar containing 3.5% P3ABA (downward triangles), 2% P3ABA (open circles) and 1% P3ABA (upward triangles). Following 15 min, 30 min and 120 min treatments, *M. tuberculosis* BSG002 cells were recovered by addition of supplemented 7H9 and incubation at 37 °C in a fresh 96-well plate. The vertical axis represents the median bioluminescence measurements (given as relative light units [RLU] per second) from recovered cells grown for approximately 21 days. The dashed line denotes background luminescence of uninoculated wells.

## Antibacterial action of surface incorporated PANI and P3ABA against the clinical isolate *M. tuberculosis* BSG002

*M. tuberculosis* BSG002 was tested using the surface decontamination assay to ascertain the activity of PANI and P3ABA against a clinical isolate. Similar to the activity against *M. tuberculosis* BSG001, both 10% and 8% PANI incorporated into agar did not mediate any meaningful knockdown of *M. tuberculosis* BSG002 (Fig. 3A).

As for *M. smegmatis* BSG200 and *M. tuberculosis* BSG001, 3.5% and 2% P3ABA agar reduced bioluminescence levels from *M. tuberculosis* BSG002 to that of the uninoculated agar (Fig. 3B). Surface incorporated 1% P3ABA knocked *M. tuberculosis* BSG002 cells

down to slightly above background levels after a 30 min treatment (Fig. 3B). Treatment of *M. tuberculosis* BSG002 with surface incorporated 3.5% P3ABA was statistically significant (Friedman test, *P* value: 0.0042). Dunn's multiple comparison test confirmed the significant activity of 3.5% P3ABA in a 15 min treatment time.

## DISCUSSION

PANI and P3ABA have previously been shown to have broad spectrum activity against a range of gram-positive and gram-negative bacteria (*Gizdavic-Nikolaidis et al., 2011a*; *Shi et al., 2006*). The results presented here demonstrate the previously untested activity of these compounds against mycobacteria. Only a 15 min exposure to 2% P3ABA was required to mediate knockdown of *M. smegmatis* and *M. tuberculosis*. This time interval was the shortest that could be tested due to the time constraints involved in the local Standard Operating Procedures for *M. tuberculosis* laboratory work. It is possible that P3ABA can decontaminate mycobacteria laden surfaces in less than a 15 min exposure.

The WHO recommends that HCWs in high incidence TB areas wear respirators to protect against TB transmission (*WHO, 2015*). More than 50% of new MDR-TB cases occur within hospitals and communities among people that haven't been previously treated for TB, which highlights the lack of adequate infection control measures in high MDR-TB incidence areas (*Menon, 2013*; *WHO, 2014*; *Kompala, Shenoi & Friedland, 2013*). Improved TB infection control is essential to curtail the development and spread of drug resistance (*Menon, 2013*). There are not many studies that address the efficacy of infection control measures on reducing transmission; however, there is limited evidence which suggests that the incidence of TB infection decreases after execution of control measures (*Joshi et al., 2006*).

Respirators are intended to protect the wearer from inhaling infectious particles by filtering small infectious droplets (*Coia et al., 2013*; *Cleveland, Robison & Panlilio, 2009*). This type of PPE is effective at preventing inhalation of infectious particles; however, the cost of the product renders respirators unobtainable for healthcare facilities in LMICs, which decreases the level of protection offered (*Menon, 2013*; *Coia et al., 2013*). Efforts to curtail the spread of MDR-TB is hampered by insufficient funding (*WHO, 2015*). The funding gaps associated with a full response to the global TB epidemic in LMICs has been steadily increasing, amounting to US$ 1.4 billion in 2015 (*WHO, 2015*).

Facemasks are worn over nose and mouth to establish a barrier between the respiratory tract and splashes and droplets in the external environment (*Coia et al., 2013*; *Cleveland, Robison & Panlilio, 2009*). Facemasks offer less protection against *M. tuberculosis* transmission as they have a lower filtration efficiency and a higher degree of face seal leakage than respirators (*Coia et al., 2013*; *Cleveland, Robison & Panlilio, 2009*; *Macintyre et al., 2014*). In a resource-limited setting where there are few/no respirators available, wearing of facemasks may help decrease transmission even if each individual HCW is not completely protected (*Menon, 2013*; *Macintyre et al., 2014*; *Nicas, 1995*). Use of a facemask that reduces inhalation of particles by 50% has been estimated to give the same level of protection as a doubling of room ventilation, but at a much lower cost (*Menon,*

*2013*). The greater affordability of facemasks is associated with fewer availability issues than respirators (*Menon, 2013*).

Facemasks provide important protection in resource-limited settings where recommended infection control measures are not feasible. However, healthcare facilities in these areas are not able to reliably provide HCWs with sterile facemasks, increasing the risk of TB transmission in high bacterial load situations, such as an infectious patient in a poorly ventilated room (*Luksamijarulkul, Aiempradit & Vatanasomboon, 2014*; *Brosseau, McCullough & Vesley, 1997*; *Rengasamy, Zhuang & Berryann, 2004*). Used contaminated facemasks can be a source of infection either due to penetration of particles through the facemask into the respiratory tract of the wearer or release of aerosols back into the air (*Luksamijarulkul, Aiempradit & Vatanasomboon, 2014*; *Brosseau, McCullough & Vesley, 1997*). Used facemasks isolated from a hospital have been shown to have bacterial contamination on the outside of the mask (*Luksamijarulkul, Aiempradit & Vatanasomboon, 2014*). *M. abscessus*, used as a surrogate for *M. tuberculosis*, is able to survive on respiratory PPE for 5 days and can be transferred to gloves during handling (*Brosseau, McCullough & Vesley, 1997*). Decontamination of facemasks by HCWs may also cause reaerosolisation of TB bacilli (*Rengasamy, Zhuang & Berryann, 2004*). A facemask impregnated with an antimicrobial agent would ameliorate the risk of TB transmission associated with these non-recommended practices. Bacteria that come into contact with the facemask surface would be killed, facilitating safer reuse and disposal as well as eliminating the need to decontaminate (*Chellamani, Veerasubramanian & Balaji, 2013*).

There are several commercially available decontaminating facemasks; however, there is limited published work on the efficacy of these facemasks. Two notable examples include silver-based facemasks and quaternary ammonium compound (QAC) based facemasks (*Li et al., 2006*; *Tseng, Pan & Chang, 2016*). The cell wall of *M. tuberculosis* is protective against penetration of silver nanoparticles into the cytoplasm as disruption with chloroform was required for inhibitory activity (*Praba et al., 2013*). *Mycobacterium avium*, a pathogenic mycobacterium, was reduced slightly in number after a 24–48 h treatment with silver nanoparticles, which suggests that the bactericidal activity of silver nanoparticles against *M. tuberculosis* in a relevant time frame (15 min) would be negligible (*Islam et al., 2013*; *Miyamoto, Yamaguchi & Sasatsu, 2000*). Furthermore, the potential for development of resistance to silver and cross-resistance to antibiotics reduces the suitability of silver as an antimycobacterial agent. Following a single exposure to silver nanoparticles, *M. smegmatis* developed resistance to silver nanoparticles, silver nitrate and the antibiotic isoniazid (*Larimer et al., 2014*). QACs have been demonstrated to be active against hydrophilic bacteria that have a negatively charged cell surface, including LPS-expressing gram-negative bacteria and teichoic-acid containing gram-positive bacteria; however, the efficacy against hydrophobic mycobacteria is limited (*Russell, 1996*; *Gottenbos et al., 2002*).

Our findings presented here identify P3ABA as a suitable candidate for the production of antimicrobial surfaces in the patient environment, including facemask materials due to the low cost of synthesis, environmental and thermal stability, and lack of mammalian cytotoxicity (*Dhand et al., 2011*; *Gizdavic-Nikolaidis et al., 2011b*). The broad spectrum

activity of fPANI may confer protection against other respiratory pathogens that also have a high burden in LMICs, such as bacterially derived pneumonia (*Macintyre et al., 2014*).

Challenges remain for researchers; the first is to demonstrate tuberculocidal activity of P3ABA in facemask material. We believe this is likely as melt blends of PANI in low density polyethylene (*Nand et al., 2013*) and P3ABA electrospun in nanofibers of poly(lactic acid) (*Gizdavic-Nikolaidis et al., 2011b*; *Gizdavic-Nikolaidis et al., 2010b*) retain antimicrobial activity. Assuming an effective facemask can be fabricated it will be important to establish that it is effective in reducing TB transmission and to determine the expected usage lifetime to give guidelines as to when to discard a used mask. As users may be tempted to reuse masks, irrespective of mask age, for the same reasons that they currently reuse infected masks; thought will still need to be given to removing the reasons for extended use and to enforcing expiration guidelines.

## CONCLUSION

A surface self-decontamination assay was established and used to test the activity of two novel antimicrobial agents, PANI and P3ABA, against mycobacteria. PANI is active against *M. smegmatis* BSG200 after 120 min exposure; however, it did not mediate surface self-sanitisation of *M. tuberculosis* BSG001 or *M. tuberculosis* BSG002. A 15 min exposure to a surface containing 3.5% P3ABA was sufficient to kill all three strains tested. Therefore, P3ABA has potential to be used to create anti-tubercular facemasks, which would serve as a cost-effective TB infection control measure in low-resource, high TB burden areas.

## ACKNOWLEDGEMENTS

The authors thank Sudip Ray, Adeline Le Cocq, Chris Wilcox and Walt Wheelwright for purified PANI and P3ABA.

### Funding

The authors received research funding from both the New Zealand Ministry of Business, Innovation and Employment (MBIE) for research programmes UOAX0812 and UOAX1410, and the University of Auckland's Vice Chancellors Strategic Development Fund, grant number 23563. The funders had no role in study design, data collection and analysis, decision to publish, or preparation of the manuscript.

### Grant Disclosures

The following grant information was disclosed by the authors:
New Zealand Ministry of Business, Innovation and Employment (MBIE) for research programmes: UOAX0812, UOAX1410.
University of Auckland's Vice Chancellors Strategic Development Fund: 23563.

### Competing Interests

Siouxsie Wiles is an Academic Editor for PeerJ.

## Author Contributions

- Julia Robertson conceived and designed the experiments, performed the experiments, analyzed the data, wrote the paper, prepared figures and/or tables, reviewed drafts of the paper.
- James Dalton conceived and designed the experiments, performed the experiments, analyzed the data, contributed reagents/materials/analysis tools, wrote the paper, reviewed drafts of the paper.
- Siouxsie Wiles conceived and designed the experiments, analyzed the data, contributed reagents/materials/analysis tools, wrote the paper, reviewed drafts of the paper.
- Marija Gizdavic-Nikolaidis conceived and designed the experiments, contributed reagents/materials/analysis tools, wrote the paper, reviewed drafts of the paper.
- Simon Swift conceived and designed the experiments, analyzed the data, wrote the paper, reviewed drafts of the paper.

## Ethics

The following information was supplied relating to ethical approvals (i.e., approving body and any reference numbers):

The University of Auckland Institutional Biological Safety Committee approved the construction and use of genetically modified risk group 2 mycobacteria (GMO11-UA007). The New Zealand Environmental Protection Agency approved the construction and use of genetically modified risk group 3 *Mycobacterium tuberculosis* (APP201346).

## Data Availability

The raw data has been supplied as a Data S1.

## Supplemental Information

Supplemental information for this article can be found online at http://dx.doi.org/10.7717/peerj.2795#supplemental-information.

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
