# Peer review of "The tuberculocidal activity of polyaniline and functionalised polyanilines"

_PeerJ, doi:10.7717/peerj.2795_

## Round 0.1 · original submission · Major Revisions

In private notes to me, the reviewers expressed these concerns:
1. Before acceptance, I would like to see the authors clarify the connection between conduction of conducting polymers and anti bacterial activity. Is it that conducting polymers are just sexy, or do they bring something new to the table?

2. Also I would like to know why there would be any difference between the timely disposal of masks currently in use versus the timely disposal of conducting polymer masks. If mask users are negligent on both accounts, then what's the point?

3. the introduction and discussion needs to focus more on the work that has been done and less on the potential applications which has not been tested.

If you can please address, thank you

Reviewer 1 ·

Basic reporting

.

Experimental design

.

Validity of the findings

.

Additional comments

1. What is the expected mask usage lifetime?
How do we know when to discard a used mask?
How will usage expiration be specified/ enforced?

2. Will users not be tempted to reuse masks irrespective of mask age for the same reason that they currently reuse infected masks?

3. The concept of “conducting polymers is important enough to be mentioned in the title but I was unable to locate anything about the correlation between polymer conductivity and anti TB effect of mask material? Specifically, Figs 1, 2, and 3 show that P3ABA mask material has a higher anti bacterial effect than PANI. How do the polymers differ in electrical conductivity? Any generalizations possible on this point?

4. Lines 512-51- “[…]cells were exposed to surface bound 3.5% P3ABA[…].” QUERY: “surface bound” how? How were the polymers fabricated into mask fabric? - by derivatizing an inert scaffold? by spinning a fiber composed of conducting monomer?

5. Text dealing with administrative matters is excessive. The ms contains too much non-science information; Pages 3 and 4 can be condensed down to as many sentences.

6.Why not present the data in Figs 1, 2, 3 in table format?

Reviewer 2 ·

Basic reporting

This is an interesting and scientifically credible and robust paper reporting a suitably detailed study of the tuberculocidal activity of conducting polymers with a view to their potential use in facemasks to improve infection control. The research focuses on determining the antibacterial effectiveness of the intrinsically conducting polymers polyaniline (PANI) and a substituted PANI, poly-3-aminobenzoic acid (P3ABA) against the microbes M. smegmatis BSG200 and M. Tuberculosis BSG001. This antibacterial effectiveness study is carried out very well using a conventional laboratory approach and scale and the results appear valid and credible.
Only PANI is specifically mentioned in the abstract. P3ABA should also be included.
The introduction and a considerable amount of the discussion is directed towards the issue of facemasks and the potential incorporation and use of PANI and P3ABA into such facemasks to likely reduce tuberculosis infection. However the experimental work has not addressed the actual incorporation of PANI and P3ABA into the facemask fabric or testing the antibacterial effectiveness of the treated facemasks. A process of incorporating these two conducting polymers into the facemask fabric and testing the antibacterial effectiveness of the treated facemask against M. smegmatis BSG200 and M. Tuberculosis BSG001 needs to be carried out either as part of this study and reported in this paper, or in a following paper. The authors mention the difficulty in processing these conducting polymers so it is likely this incorporation would pose a scientific and technological challenge. Also, if the polymers are incorporated into the fibres of the facemask fabric rather than coating them, their antibacterial effectiveness may be compromised. There may be some consumer resistance to black or very dark coloured facemasks due to the conducting polymer components.
In the discussion, the authors make the assumption that because they have shown PANI and P3ABA to be effective against M. smegmatis BSG200 and M. Tuberculosis BSG001 microbes in laboratory tests that they will be equally effective in facemasks. This also needs to be demonstrated either here or in a follow up paper. Hence the extent of the narrative on facemasks in both the introduction and the discussion should be curtailed and the focus should be on the successful laboratory scale demonstration of the effectiveness of the “pure” PANI and P3ABA conducting polymers against the M. smegmatis BSG200 and M. Tuberculosis BSG001 microbes, which is indeed a credible and very useful study.
Much of the information on tuberculosis and facemasks contained in the discussion section would be better presented in a more curtailed form, in the introduction section.

Experimental design

The experimental work regarding the testing of the laboratory scale standard testing of the effectiveness of the “pure” PANI and P3ABA conducting polymers against the M. smegmatis BSG200 and M. Tuberculosis BSG001 microbes, has been very well carried out and the results produced are scientifically credible.
The reason why genetically modified risk group 2 mycobacteria(GMO11-UA007) was used is unclear. This should be explained further.
In the statistical analysis section the exposure time should be related to a realistic timeframe for the required knockdown to be achieved on the facemask when in everyday use.

Validity of the findings

As mentioned above the findings for the standard laboratory scale testing of the antibacterial effectiveness of the conducting polymers PANI and P3ABA conducting polymers against the M. smegmatis BSG200 and M. Tuberculosis BSG001 microbes, are valid. However without testing on actual facemask fabric, these findings cannot be unequivocally extended to such fabrics.

Additional comments

The comments are contained in the boxes above.

---

## Round 0.2 · accepted · Accept

Examining the reviews, it appears that Reviewer 1 is suggesting rejection based on lack of novelty (they note that antimicrobial activity has already been shown, hence their implication is that there is nothing new here). Lack of novelty is not one of our criteria (we allow replication etc experiments). This experiment was testing the concentrations etc of P3ABA that might be required

Therefore, we are happy to accept.

Reviewer 1 ·

Basic reporting

The anti-mycobacterial activity of the polyanilines, including fabrication into nanofibers that have antimicrobial activity was previously established as noted in the ms. The present studies would would fbest be included in a report of the actual fabrication and testing the effectiveness of a mask. That would be the most effective validation of the polymer-infused agar as a surrogate for an actual working mask model.

Experimental design

See above.

Validity of the findings

See above

Additional comments

Wet agar is not validated as a model for a porous dry woven cloth mask.

Reviewer 2 ·

Basic reporting

The issues I raised in my original review of this paper have been addressed satisfactorily.

Experimental design

This is sound and meets the journal standard.

Validity of the findings

The experimental work has been carried out rigorously and the findings are valid.

Additional comments

This is a very good paper now. The experimental work has been carried out in a careful and logical manner and the results obtained appear valid. The results have been interpreted and discussed appropriately as has been their relevance to the potential use of the conducting polymers, particularly P3ABA in face masks to limit cross infection of TB.
The paper should now be published.